# Assessment of Saudi Public Perceptions and Opinions towards Artificial Intelligence in Health Care

**DOI:** 10.3390/medicina60060938

**Published:** 2024-06-04

**Authors:** Wajid Syed, Salmeen D. Babelghaith, Mohamed N. Al-Arifi

**Affiliations:** Department of Clinical Pharmacy, College of Pharmacy, King Saud University, Riyadh 11451, Saudi Arabia; sbabelghaith@ksu.edu.sa (S.D.B.); malarifi@ksu.edu.sa (M.N.A.-A.)

**Keywords:** perceptions, opinion, artificial intelligence, Saudi public, advancing health

## Abstract

*Background and Objectives:* The healthcare system in Saudi Arabia is growing rapidly with the utilization of advanced technologies. Therefore, this study aimed to assess the Saudi public perceptions and opinions towards artificial intelligence (AI) in health care. *Materials and Methods:* This cross-sectional web-based questionnaire study was conducted between January and April 2024. Data were analyzed from 830 participants. The perceptions of the public towards AI were assessed using 21-item questionnaires. *Results:* Among the respondents, 69.4% were males and 46% of them were aged above 41 years old. A total of 84.1% of the participants knew about AI, while 61.1% of them believed that AI is a tool that helps healthcare professionals, and 12.5% of them thought that AI may replace the physician, pharmacist, or nurse in the healthcare system. With regard to opinion on the widespread use of AI, 45.8% of the study population believed that healthcare professionals will be improved with the widespread use of artificial intelligence. The mean perception score of AI among males was 38.4 (SD = 6.1) and this was found to be higher than for females at 37.7 (SD = 5.3); however, no significant difference was observed (*p* = 0.072). Similarly, the mean perception score was higher among young adults aged between 20 and 25 years at 38.9 (SD = 6.1) compared to other age groups, but indicating no significant association between them (*p* = 0.198). *Conclusions:* The results showed that the Saudi public had a favorable opinion and perceptions of AI in health care. This suggests that health management recommendations should be made regarding how to successfully integrate and use medical AI while maintaining patient safety.

## 1. Introduction

The healthcare system has developed enormously everywhere, more particularly in the developed countries, due to the adoption of advanced technologies such as artificial intelligence (AI). The artificial intelligence system has received a great deal of attention as a result of the utilization of cutting-edge technologies to deliver the greatest caliber of health care [1,2,3]. The theory and development of computer systems that can do activities that ordinarily require human intellect, such as speech recognition, visual perception, decision-making, and language translation, are referred to as artificial intelligence [4]. This demonstrates that AI functions similarly to humans, but not identically; this function is currently being developed [5,6]. Artificial intelligence is revolutionizing data science and information technology by advancing automated tasking technologies [4,5,6].

The utilization of artificial intelligence has been increasing in various healthcare sectors and industries with multiple benefits in terms of rapid healthcare delivery and reduced work time and overload [4,5,6,7,8,9]. For instance, the literature revealed that artificial intelligence helps healthcare professionals in terms of patient record keeping and in the management of medical diagnosis and treatment of various diseases, in addition to AI improving human decision-making and efficiency [4,5]. According to a prior study by Michael Gerlich, the increasing adoption of AI in daily life is expected to lead to increased productivity and creativity [8]. The cornerstone for AI’s broad adoption is the European Commission’s set of trustworthy AI principles, which emphasize data protection, security, and wise governance [8]. In a similar vein, Yeh et al.’s 2021 survey of Taiwanese citizens in China revealed that they have a significant amount of trust in their understanding of AI. Although they believed AI to be unsafe, they also had a highly optimistic attitude toward it [9]. Taiwanese people may be “rational optimists” in general when it comes to AI [9].

Several national and internal surveys by the public and students have found positive views of artificial intelligence in health care [4,10,11]. One study, for instance, found that students were generally well informed about artificial intelligence (AI) and that most respondents had favorable opinions of the technology’s principles, applications, and advantages [4]. Similarly, a study conducted among the public found that AI may also improve the efficacy and accuracy of risk forecasts, which would improve public health outcomes [10]. Furthermore, prior research indicated that AI aids in machine learning algorithms that can monitor vast volumes of data, such as electronic health records, to identify trends and forecast the chance of illnesses [10].

The public literature on AI in Saudi Arabia and other countries is scarce [12,13]. For example, a recent survey by Ipsos [12] evaluating public opinion toward artificial intelligence in Saudi Arabia found that 77% of Saudis knew what artificial intelligence (AI) was. Furthermore, when it comes to perceptions of AI, eight out of ten people are enthusiastic about goods and services that use the technology [12]. The majority of Saudi citizens also believe that AI will improve their quality of life in various aspects [12]. People’s perceptions significantly impact results and implementation [12,13]. Furthermore, there is a dearth of literature on the topic. Therefore, this study aims to assess Saudi public perceptions and opinions towards artificial intelligence (AI) in health care.

## 2. Methodology

### 2.1. Study Design, Setting, and Population

We conducted a cross-sectional, self-completed questionnaire using convenience sampling, among the public living in Saudi Arabia, over 18 weeks from 1 January 2024, to 1 April 2024. We recruited individuals living in Saudi Arabia. Eligible individuals were over the age of 18 years and able to understand the local language (Arabic) and the questionnaires. The study excluded those who did not meet the inclusion criteria. The King Saud University Research Ethics Committee at the College of Medicine in Riyadh, Saudi Arabia, approved the study. In addition, this study followed the guidelines of the Declaration of Helsinki for human research. Prior to administering the study questionnaires, we informed the participants that we would solely use the data for research purposes, maintain confidentiality throughout the investigation, and provide them with the option to withdraw from the study at any time. Figure 1 provides a step-by-step breakdown of the study design.

### 2.2. Sample Size Estimation

We calculated the sample size for this study using the online calculator, taking into account a population of 7,821,000 in Riyadh, Saudi Arabia in 2024, with a 5% margin of error (ME) and a 95% confidence interval (CI). We estimated the required sample to be 388, but we approached 1000 respondents to account for non-response and reduce data collection errors.

### 2.3. Questionnaire Design

We designed a 21-item self-prepared and pretested survey in the English language to achieve the study objectives, drawing on the previous literature [4]. We divided the total items into four parts, with part one collecting demographic information on the respondent’s age, gender, and educational level. Six items in the second section collected data on the public’s opinions about artificial intelligence. We used a total of 12 items in the third section to gather data regarding the respondents’ perceptions about AI. A team of experts, including a researcher from the clinical pharmacy department and a professor, reviewed the initial draft of the questionnaires. Later, a native Arabic speaker assisted in translating the questionnaires into Arabic using forward and backward translation procedures. We conducted a pilot study among randomly selected individuals (*n* = 20) from Riyadh to determine the validity and reliability. The main results did not incorporate the findings of the pilot study, and the questionnaires remained unchanged. We determined the reliability of the questionnaires using Cronbach’s alpha, and found this to be 0.80, indicating their validity for use in the study. A group leader from each college assisted in collecting the data using a convenience sampling procedure.

We collected the data from individuals living in the Riyadh region of Saudi Arabia. We prepared electronic questionnaires using Google Forms for data collection and sent them to the targeted population via social media. In addition, before beginning the study using questionnaires in the Google form, there was a statement about the significance and confidentiality of the data; those who agreed to proceed would be redirected to the main questionnaires, and this was considered informed consent. We followed up with the participants to maximize the number of responses, reminding them to complete the survey and return it. Researchers from the College of Pharmacy’s Clinical Pharmacy Department collected the data using online questionnaires. We used a total of twelve questions to evaluate participants’ perceptions of AI. We scored all the questions on a five-point Likert scale (strongly agree, agree, neutral, disagree, and strongly disagree). The total score was calculated. For the positive questions, the score on the Likert scale was 1 (strongly disagree) to 5 (strongly agree), and for the negative questions, the score was the reverse of the positive questions.

## 3. Data Analysis

We performed data analysis using the SPSS statistical software package, version 26 (SPSS Inc., Armonk, NY, USA). Descriptive analyses such as frequencies (*n*), percentages (%), mean, and standard deviations (Std) were used, and to find out the association between variables, the ANOVA and Student’s *t*-test were used at a level of *p* < 0.05, which was considered statistically significant.

## 4. Results

A total of 830 individuals completed and responded to the study questionnaires, giving a response rate of 83% (*n* = 830). In this study, the majority of the respondents were male (69.4%), while 30.6% of them were female, 46% of them were above 41 years old, and 20.8% of them were between the ages of 19 and 25 years old, as shown in Table 1. Furthermore, in terms of education, the majority of them belonged to universities (79.3%), while 17.7% were in high school. On the other hand, most of them were Saudi nationals. Table 1 presents the demographic characteristics.

In this study, 84.1% of the population knew about AI, while 61.1% (*n* = 507) of the population believed that AI is a tool that helps healthcare professionals (Figure 2); furthermore, 12.5% (*n* = 104) of them thought that artificial intelligence may replace the physician, pharmacist, or nurse in the healthcare system, and 5.3% (*n* = 127) of them disagreed about it, as shown in Figure 2.

Regarding the widespread use of artificial intelligence, 45.8% of the study population believed that it would more effectively equip healthcare professionals. Table 2 provides the detailed frequencies of the population’s opinions about artificial intelligence.

In this study, one third of the respondents disagreed that AI devalues the medical profession, while half of the respondents agreed or strongly agreed that AI reduces errors in medical practice. Furthermore, the majority (80.1%) of the respondents agreed that AI facilitates healthcare professionals’ access to information; on the other hand, 70.4% of the respondents agreed that AI enables healthcare professionals to make more accurate decisions. Table 3 provides the detailed frequencies of the individuals’ perceptions of AI in health care.

The mean perception score of AI among individuals was found to have no significant impact; however, the mean perception score of AI among males (38.4, SD = 6.1) was found to be higher than for females (37.7, SD = 5.3) with borderline significance (*p* = 0.072). On the other hand, the mean perception score was higher among young adults (38.9, SD = 6.1) compared to other age groups, but indicating no significant association between them (*p* = 0.198). Similarly, the education of the respondents did not have any significant impact on the mean perception score of AI, as shown in Table 4.

## 5. Discussion

The present study aimed to evaluate the Saudi public’s perceptions and opinions towards artificial intelligence in health care. We identified a limited amount of literature on public perceptions of artificial intelligence in health care [5]. However, the majority of this literature came from healthcare students [4]. This study will significantly raise awareness among individuals and healthcare professionals about how this perception contributes to the advancement of artificial intelligence, thereby improving the quality of care for patients and community members, and in hospitals, and clinical environments. Furthermore, this study’s findings will serve as a reference for much-needed future studies.

According to our results, 84.1% of the individuals knew about AI, while 61.1% of them revealed that artificial intelligence is a tool that helps healthcare professionals. Furthermore, 45.8% of the public believed that healthcare professionals would be better off with the widespread use of artificial intelligence; on the other hand, 24.7% of Saudis fear losing jobs with the introduction of artificial intelligence. These findings were similar to those of earlier studies published among healthcare students and the public [4,12,13]. For instance, Liehner et al. assessed the public’s perceptions, attitudes, and trust towards artificial intelligence and reported that AI is a tool that can act independently, adapt, and help them in their daily lives. In addition, earlier findings also revealed that respondents are willing to use AI; nevertheless, they are less willing to place their trust in the technology [12]. Similarly, a study conducted among healthcare students yielded similar results, revealing that artificial intelligence serves as a valuable tool for healthcare professionals. While approximately 25% of the students agreed that the introduction of artificial intelligence could lead to job losses, 58% of them believed that the widespread use of artificial intelligence would benefit healthcare professionals. [4].

The current findings revealed a positive perception of AI in health care among respondents who disagreed that AI devalues the medical profession, while half of the respondents agreed or strongly agreed that AI reduces errors in medical practice. Furthermore, the majority (80.1%) of the respondents agreed that AI facilitates healthcare professionals’ access to information; on the other hand, 70.4% of the respondents agreed that AI enables healthcare professionals to make more accurate decisions. These findings were comparable to earlier findings by Ipsos, where the author reported that the majority of the Saudi public has a good understanding of AI, and the majority also believe AI will shortly enhance various aspects of life, including the economy in Saudi Arabia (76%), time management (72%), entertainment options (69%), and various other areas [13]. Another study among the public reveals that people’s appreciation of AI’s advantages and potential, along with their concerns, anxieties, and worries about this enigmatic and mysterious technology, frequently influence its use [14]. Although public perceptions of AI in health care are limited, a study examined the public perceptions of AI in defense and reported several false beliefs on the uses of AI in safety, many of which hold that a wide range of AI-related technologies are already in use [15]. This demonstrated how credible sources of information might coexist with conspiracy theories and stories from the media [15]. The report elucidates the misconceptions and information gaps that require resolution, and offers valuable guidance on the reliable, effective, and sufficient dissemination of information to the public regarding the benefits, drawbacks, and hazards of artificial intelligence in everyday life and healthcare environments. [15].

In this study, the mean score of AI was not significantly associated with gender, age, or educational level. Meanwhile, previous studies among healthcare professionals showed age, profession, and working experience had a significant impact on the perception score [11], In addition, earlier findings revealed no significant association between gender, nationality, and Saudi regions in terms of perception score [11]. Naturally, healthcare professionals undergo some form of training and possess AI experience in comparison to the general public [11]. This study would make a significant contribution to AI implementation and would serve as a reference for future studies. Educational and healthcare institutions could utilize the findings to create suitable training programs that enhance AI knowledge and awareness in health care. Additionally, research indicates that artificial intelligence (AI) may enhance the capacity of public health to advance everyone’s health in every community [16,17]. Public health organizations must carefully consider their AI implementation techniques in order to properly fulfill this promise and apply AI to public health tasks [18,19,20].

Despite its benefits in health care, AI has ethical concerns such as data privacy, algorithmic bias, and the potential impact of AI on the patient–provider relationship [21,22,23,24]. For example, healthcare decisions may be biased as a result of the data used to train AI algorithms [21,22,23,24]. This could lead to ethical dilemmas, as AI could potentially exacerbate or sustain disparities in healthcare outcomes among specific demographic groups [21,22,23,24]. Furthermore, due to a lack of heterogeneity in data representation, questions about data sharing and triangulation are emerging with the growing use of AI in healthcare specialties [21,22,23,24]. We must enforce federal regulations pertaining to the sharing and use of health data, as a data breach can have consequentialist or deontological consequences, or both [21,22,23,24]. AI models that preserve patient privacy and properly address safety concerns include federated learning, differential privacy, and cryptography approaches [21,22,23,24].

The current study has several limitations. First, the data presented here were limited to respondents from the Riyadh region, which may not fully represent the diversity of opinions across all regions of Saudi Arabia. Expanding the geographical scope would enhance the generalizability of the findings. Due to the self-administered nature of the questionnaire used to gather the data, recall or social desirability bias may have been present. Lastly, the convenience sampling procedure used might be another limitation to this study. In addition, the age and gender distribution in the current study did not exactly match the number used as references for sample size calculation, which suggests that future research with a larger sample size, including data from other regions of Saudi Arabia, is needed to better corroborate the findings. In addition to this, the lack of qualitative insights is one of the limitations; while the quantitative approach provides valuable statistical data, incorporating qualitative insights through interviews or open-ended questions could offer a deeper understanding of the reasons behind the public’s perceptions and opinions. Despite these limitations, the authors believe that the study provides useful insights into individual’s perspectives on the integration and use of AI in healthcare facilities in Saudi Arabia.

## 6. Conclusions

The results showed that the Saudi public had a favorable opinion and perception of AI in health care. We propose providing health management guidelines and insights on the effective integration and application of medical AI, all while ensuring patient safety. Moreover, the conclusions could highlight and reference specific findings related to the lack of understanding and knowledge of HDIs among nursing students, thereby strengthening the argument. Furthermore, incorporating HM and HDI courses into the academic curriculum can effectively educate and teach undergraduates about HMs and HDIs. This could help graduates overcome obstacles on the practice site.

## Figures and Tables

**Figure 1 medicina-60-00938-f001:**
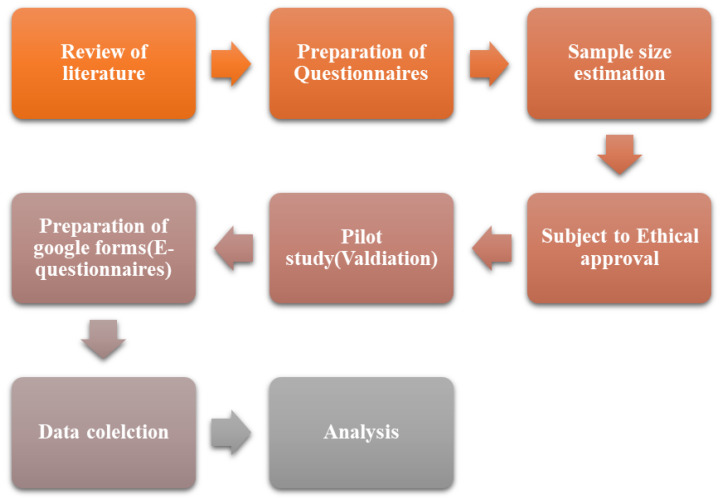
Flow diagram of the study.

**Figure 2 medicina-60-00938-f002:**
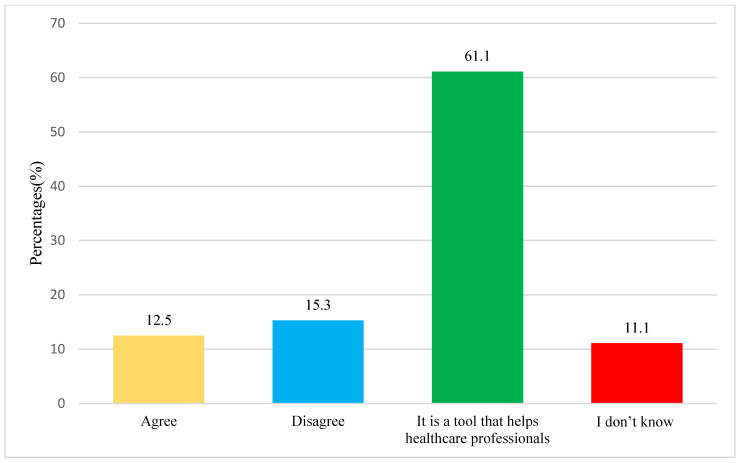
Opinions of the widespread use of artificial intelligence.

**Table 1 medicina-60-00938-t001:** Respondent’s demographic data (*n* = 830).

Variables	Number (*n*)	Percentage (%)
Gender		
Male	575	69.4
Female	254	30.6
Age		
19–25 years	173	20.8
26–30 years	63	7.6
31–35 years	67	8.1
36–40 years	145	17.5
>41 years	382	46.0
Education level		
Less than high school	25	3.0
High school	147	17.7
University	658	79.3
Nationality		
Saudi	785	94.6
Non-Saudi	45	5.4

**Table 2 medicina-60-00938-t002:** Opinions of the public about artificial intelligence (AI) (*n* = 830).

Variables	Number (*n*)	Percentage (%)
Knowledge of AI		
Yes	698	84.1
No	132	15.9
What is your opinion, if artificial intelligence is widespread in Saudi Arabia?		
Risk of losing jobs with the introduction of robots (artificial intelligence) with the decrease in the need for employees	205	24.7
Healthcare professionals will be better with the widespread use of artificial intelligence.	380	45.8
The choice of specialization field will be influenced by how artificial intelligence is used in that field	145	17.5
I don’t know	100	12.0
Have you received any formal education?		
about artificial intelligence?		
Yes	62	7.5
No	675	81.3
Received some information over the internet	37	4.5
Through friends	56	6.7
Do you know ChatGPT		
Yes	412	49.6
No	418	50.4
Have you used ChatGPT?		
Yes	295	35.5
No	353	64.5

**Table 3 medicina-60-00938-t003:** Perceptions of the respondents about AI (*n* = 380).

Variables	Strongly Agree *n* (%)	Agree *n* (%)	Neutral *n* (%)	Disagree *n* (%)	Strongly Disagree *n* (%)
AI devalues the medical profession	83 (10)	120 (14.5)	297 (35.8)	252 (30.4)	78 (9.4)
AI reduces errors in medical practice	147 (17.70)	275 (33.1)	270 (32.5)	114 (13.7)	24 (29.0)
AI facilitates patients’ access to the service	213 (25.7)	411 (49.5)	159 (19.2)	38 (4.6)	9 (1.1)
AI facilitates healthcare professionals’ access to information	275 (33.1)	397 (47.8)	127 (15.3)	22 (2.7)	9 (1.1)
AI enables healthcare professionals to make more accurate decisions	228 (27.5)	356 (42.9)	190 (22.9)	38 (4.6)	18 (2.2)
AI increases patients’ confidence in medicine	148 (17.8)	283 (34.1)	263 (31.7)	101 (12.2)	35 (4.2)
AI facilitates patient education	171 (20.6)	387 (46.6)	203 (24.5)	51 (6.1)	18 (2.2)
AI negatively affects the relationship between healthcare professionals and the patient	92 (11.1)	174 (21.0)	298 (35.9)	221 (26.6)	45 (5.4)
AI damages the trust that is the basis of the healthcare professional’s relationship	107 (12.9)	196 (23.6)	272 (32.8)	205 (24.7)	50 (6.0)
AI reduces the humanistic aspect of the medical profession.	170 (20.5)	232 (28.0)	201 (24.2)	179 (21.6)	48 (5.8)
AI violations of professional confidentiality may occur more often	176 (21.2)	246 (29.6)	248 (29.9)	120 (14.5)	40 (4.8)
AI allows the patient to increase their control over their health	146 (17.6)	309 (37.2)	268 (32.3)	89 (10.7)	18 (2.2)

**Table 4 medicina-60-00938-t004:** The variance between the mean perception score of AI and the characteristics of the participants.

Variables	Mean	SD	*f*-Value	*t*-Value	*p*-Value
Gender					
Male	38.4	6.1	3.282	1.463	0.072 *
Female	37.7	5.3			
Age					
20–25	38.9	6.1	1.507	--	0.198 **
26–30	38.7	6.5			
31–35	38.4	5.1			
36–40	38.3	5.3			
>41	37.3	6.0			
Education levels					
Less than high school	36.4	7.6	1.161	--	0.314 **
High school	38.3	5.8			
University	38.1	5.8			

* Student’s *t*-test; ** ANOVA.

## Data Availability

The data used in this study will be available from the correspondence author upon request.

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
