# Peer review of "Assessment of Saudi Public Perceptions and Opinions towards Artificial Intelligence in Health Care"

_medicina, 2024, doi:10.3390/medicina60060938_

Round 1
Reviewer 1 Report
Comments and Suggestions for Authors
Strengths
Relevant Topic: The study addresses a timely and pertinent topic, considering the increasing importance of AI in healthcare globally. Understanding public perceptions in Saudi Arabia provides valuable insights for policymakers and healthcare providers.
Comprehensive Data Collection: The use of a 21-item questionnaire ensures a thorough assessment of public perceptions and opinions towards AI. The inclusion of demographic variables allows for detailed analysis of different subgroups within the population.
Statistical Analysis: The use of SPSS for data analysis, including ANOVA and t-tests, provides a robust statistical framework for evaluating the associations between variables and ensuring the reliability of the findings.
High Response Rate: Achieving an 83% response rate (830 out of 1000 respondents) strengthens the validity of the study's findings and indicates strong engagement from the participants.
Weaknesses
Limited Geographical Scope: The study is limited to respondents from the Riyadh region, which may not fully represent the diversity of opinions across all regions of Saudi Arabia. Expanding the geographical scope would enhance the generalizability of the findings.
Potential Bias in Sample: The convenience sampling method and the self-administered nature of the questionnaire could introduce selection bias and social desirability bias, potentially affecting the accuracy of the results.
Lack of Qualitative Insights: While the quantitative approach provides valuable statistical data, incorporating qualitative insights through interviews or open-ended questions could offer a deeper understanding of the reasons behind the public's perceptions and opinions.
Insufficient Discussion on Ethical Concerns: The study touches on ethical concerns but does not delve deeply into issues such as data privacy, algorithmic bias, and the potential impact of AI on the patient-provider relationship. A more thorough discussion of these ethical considerations would add depth to the analysis.
Detailed Comments
Introduction: The introduction effectively highlights the rapid development of AI in healthcare and sets the stage for the study. However, providing more context on the specific advancements and challenges in Saudi Arabia's healthcare system would better frame the need for this research.
Literature Review: The literature review provides a solid foundation by referencing relevant studies on public perceptions of AI in healthcare. Including more recent and region-specific studies would further strengthen this section.
Methodology: The methodology is clearly outlined, detailing the sample size calculation, questionnaire design, and data collection process. However, providing more information on the validation of the questionnaire and the pilot study results would enhance the credibility of the instrument used.
Results: The results are presented clearly, with tables and figures that aid in understanding the data. The analysis of demographic variables and their association with perception scores is thorough. However, discussing the implications of these associations in more detail would provide greater insight.
Discussion: The discussion section effectively interprets the findings and compares them with previous studies. Addressing the limitations of the study and suggesting specific strategies for overcoming them would improve this section.
Conclusion: The conclusion summarizes the key findings and their implications for healthcare policy and practice. Including recommendations for future research and practical applications of AI in healthcare would provide a more forward-looking perspective.
Comments on the Quality of English LanguageStrengths
Relevant Topic: The study addresses a timely and pertinent topic, considering the increasing importance of AI in healthcare globally. Understanding public perceptions in Saudi Arabia provides valuable insights for policymakers and healthcare providers.
Comprehensive Data Collection: The use of a 21-item questionnaire ensures a thorough assessment of public perceptions and opinions towards AI. The inclusion of demographic variables allows for detailed analysis of different subgroups within the population.
Statistical Analysis: The use of SPSS for data analysis, including ANOVA and t-tests, provides a robust statistical framework for evaluating the associations between variables and ensuring the reliability of the findings.
High Response Rate: Achieving an 83% response rate (830 out of 1000 respondents) strengthens the validity of the study's findings and indicates strong engagement from the participants.
Weaknesses
Limited Geographical Scope: The study is limited to respondents from the Riyadh region, which may not fully represent the diversity of opinions across all regions of Saudi Arabia. Expanding the geographical scope would enhance the generalizability of the findings.
Potential Bias in Sample: The convenience sampling method and the self-administered nature of the questionnaire could introduce selection bias and social desirability bias, potentially affecting the accuracy of the results.
Lack of Qualitative Insights: While the quantitative approach provides valuable statistical data, incorporating qualitative insights through interviews or open-ended questions could offer a deeper understanding of the reasons behind the public's perceptions and opinions.
Insufficient Discussion on Ethical Concerns: The study touches on ethical concerns but does not delve deeply into issues such as data privacy, algorithmic bias, and the potential impact of AI on the patient-provider relationship. A more thorough discussion of these ethical considerations would add depth to the analysis.
Detailed Comments
Introduction: The introduction effectively highlights the rapid development of AI in healthcare and sets the stage for the study. However, providing more context on the specific advancements and challenges in Saudi Arabia's healthcare system would better frame the need for this research.
Literature Review: The literature review provides a solid foundation by referencing relevant studies on public perceptions of AI in healthcare. Including more recent and region-specific studies would further strengthen this section.
Methodology: The methodology is clearly outlined, detailing the sample size calculation, questionnaire design, and data collection process. However, providing more information on the validation of the questionnaire and the pilot study results would enhance the credibility of the instrument used.
Results: The results are presented clearly, with tables and figures that aid in understanding the data. The analysis of demographic variables and their association with perception scores is thorough. However, discussing the implications of these associations in more detail would provide greater insight.
Discussion: The discussion section effectively interprets the findings and compares them with previous studies. Addressing the limitations of the study and suggesting specific strategies for overcoming them would improve this section.
Conclusion: The conclusion summarizes the key findings and their implications for healthcare policy and practice. Including recommendations for future research and practical applications of AI in healthcare would provide a more forward-looking perspective.
Author Response
Comments and Suggestions for Authors
Strengths
Relevant Topic: The study addresses a timely and pertinent topic, considering the increasing importance of AI in healthcare globally. Understanding public perceptions in Saudi Arabia provides valuable insights for policymakers and healthcare providers.
Response: thank you we agreed its significance
Comprehensive Data Collection: The use of a 21-item questionnaire ensures a thorough assessment of public perceptions and opinions towards AI. The inclusion of demographic variables allows for detailed analysis of different subgroups within the population.
Response: thank you we appreciate your comment
Statistical Analysis: The use of SPSS for data analysis, including ANOVA and t-tests, provides a robust statistical framework for evaluating the associations between variables and ensuring the reliability of the findings.
Response: thank you we appreciate your comment
High Response Rate: Achieving an 83% response rate (830 out of 1000 respondents) strengthens the validity of the study's findings and indicates strong engagement from the participants.
Response: thank you we appreciate your comment
Weaknesses
Limited Geographical Scope: The study is limited to respondents from the Riyadh region, which may not fully represent the diversity of opinions across all regions of Saudi Arabia. Expanding the geographical scope would enhance the generalizability of the findings.
Response: thank you the comment, we added it to one of the limitation
Potential Bias in Sample: The convenience sampling method and the self-administered nature of the questionnaire could introduce selection bias and social desirability bias, potentially affecting the accuracy of the results.
Response: thank you the comment, we added it to one of the limitation
Lack of Qualitative Insights: While the quantitative approach provides valuable statistical data, incorporating qualitative insights through interviews or open-ended questions could offer a deeper understanding of the reasons behind the public's perceptions and opinions.
Response: thank you the comment, we added it to one of the limitation
Insufficient Discussion on Ethical Concerns: The study touches on ethical concerns but does not delve deeply into issues such as data privacy, algorithmic bias, and the potential impact of AI on the patient-provider relationship. A more thorough discussion of these ethical considerations would add depth to the analysis.
Response: thank you the comment, we added about ethical concern of ai …as… Despite its benefits in healthcare AI, has ethical concerns such as data privacy, algorithmic bias, and the potential impact of AI on the patient-provider relationship [21-24]. For example, healthcare decisions may be prejudiced as a result of data used to train AI algorithms [21-24]. This may result in moral conundrums where AI may worsen or maintain differences in healthcare outcomes between certain demographic groups [21-24]. Furthermore, due to a lack of heterogeneity in data representation, questions about data sharing, triangulation, are emerging with the growing use of AI in healthcare specialties [21-24]. Federal regulations pertaining to the sharing and use of health data must be enforced because the consequences of a data breach can be consequentialist, deontological, or both [21-24]. AI models that preserve patient privacy and properly address safety concerns include federated learning, differential privacy, and cryptography approaches [21-24]
Detailed Comments
Introduction: The introduction effectively highlights the rapid development of AI in healthcare and sets the stage for the study. However, providing more context on the specific advancements and challenges in Saudi Arabia's healthcare system would better frame the need for this research.
Response: thank you the comment, we added and updated the introduction as… According to a prior study by Michael Gerlich, the increasing adoption of AI in daily life is expected to lead to increased productivity and creativity [8]. The cornerstone for AI's broad adoption is the European Commission's set of Trustworthy AI principles, which emphasize data protection, security, and wise governance [8]. In a similar vein, Yeh et al.'s 2021 survey of Taiwanese citizens in China revealed that they have a significant amount of trust in their understanding of AI. Although they believed AI to be unsafe, they also had a highly optimistic attitude toward it [9]. Taiwanese people may be "rational optimists" in general when it comes to AI [9].
Positive views of artificial intelligence in healthcare have been found through several national and internal surveys by the public and students [4,10,11]. One study, for instance, found that students were generally well-informed about artificial intelligence (AI) and that most respondents had favorable opinions of the technology's principles, applications, and advantages [4]. Similarly, a study among the public, reported that AI may also improve the efficacy and accuracy of risk forecasts, which would improve public health outcomes [10]. Furthermore, prior research indicated that AI aids in Machine learning algorithms can monitor vast volumes of data, such as electronic health records, to identify trends and forecast the chance of illnesses [10].
Literature Review: The literature review provides a solid foundation by referencing relevant studies on public perceptions of AI in healthcare. Including more recent and region-specific studies would further strengthen this section.
Response: thank you for the comment, there is no study about AI perceptions in ksa, however we have found one survey and we added …as……. For example, a recent survey by Ipsos survey [12] evaluating public opinion toward artificial intelligence in Saudi Arabia found that 77% of Saudis knew what artificial intelligence (AI) was. Furthermore, regarding perceptions toward AI, eight out of ten people are enthusiastic about goods and services that use the technology [12]. The majority of Saudi citizens also think AI would improve their quality of life shortly in various aspects of life [12].
Methodology: The methodology is clearly outlined, detailing the sample size calculation, questionnaire design, and data collection process. However, providing more information on the validation of the questionnaire and the pilot study results would enhance the credibility of the instrument used.
Response: thank you , here is the details… The validity and reliability were determined using a pilot study conducted among randomly selected individuals (n=20) from Riyadh. The findings of the pilot study were not included in the main results and there were no amendments were made in the questionnaires. The reliability of the questionnaires was determined using Cronbach alpha, which was found to be 0.80, indicating the questionnaires are valid to use in the study.
Results: The results are presented clearly, with tables and figures that aid in understanding the data. The analysis of demographic variables and their association with perception scores is thorough. However, discussing the implications of these associations in more detail would provide greater insight.Thank you for the comment, we have added as… In this study, the mean score of AI was not significantly associated with gender, age, or educational levels. While previous study among healthcare professionals shows age, profession, and working experience had a significant impact on the perception score [11]. In addition, earlier findings revealed no significant association between gender, nationality, and Saudi regions with a perception score [11]. This could be better explained by the fact that healthcare professionals of course undergo some kind of courses and have experience in AI, compared to the general public [11]. This study would add a significant contribution to the implementation of AI and would serve as a reference for future studies. The findings could also be used by educational and healthcare institutions to develop appropriate training to improve AI courses to
Discussion: The discussion section effectively interprets the findings and compares them with previous studies. Addressing the limitations of the study and suggesting specific strategies for overcoming them would improve this section.
Response: thank you for the comment, we appreciate it , we have added a detailed limitations as The current study has several limitations. First, the data presented here were limited to respondents from the Riyadh region, which may not fully represent the diversity of opinions across all regions of Saudi Arabia. Expanding the geographical scope would enhance the generalizability of the findings. Due to the self-administered nature of the questionnaire used to gather the data, recall or social desirability bias may have been present. Last the convenience sampling procedure, which might be one of the limitation to study. In addition, the age, and gender in the current study not exactly the number which used as references for sample size calculation, which suggest that it is recommended future research with a larger sample size, including data from other regions of Saudi Arabia is needed to better corroborate the findings. In addition to this Lack of Qualitative Insights is one of the limitations, While the quantitative approach provides valuable statistical data, incorporating qualitative insights through interviews or open-ended questions could offer a deeper understanding of the reasons behind the public's perceptions and opinions
Conclusion: The conclusion summarizes the key findings and their implications for healthcare policy and practice. Including recommendations for future research and practical applications of AI in healthcare would provide a more forward-looking perspective.
Response: thank you for the comment, we appreciate it

Reviewer 2 Report
Comments and Suggestions for Authors
Dear Authors,
Your article is very interesting! However I have several comments:
The Abstract needs some improvement. The aim of the study is to assess (not to assessment). The statistical software and methods should not be mentioned there. The number of the respondents should be moved to methods. „This is a cross-sectional web-based questionnaire study that was conducted between January to April 2024 among 830 participants of which 69.4% were males, 46% of them were aged above 41 years old“. I would suggest to avoid +- sing when reporting SD. You could use parenthesis of comma instead. The sentence „Similarly, the mean perception score was higher among young adults (38.9±6.1) compared to another age group“ is not clear. Did you mean the other age groups? And please clarify what do you mean by young (name the age group instead of describing it). Is this difference significant? I would suggest to add the p-values to all comparisons.
The conclusion seems to consist of one sentence but strangely separated. Please rewrite it.
Methodology
„Eligible individuals were over the age of >18 years“ – did you mean 18 and above? Please provide more details how did you ensure the participation of different types of individuals.
I would also suggest to restructure the section: describe the study population; the sample size calculation; the respondents‘ selection; the questionnaire.
„The sample size for this study was calculated using the online calculator by considering an unknown (n=20,000) population of Riyadh in Saudi Arabia, at a 5% margin of error (ME) and a 95% confidence interval (CI). required sample was estimated to be 388, however to account for nonresponse and reduce data collecting errors we approached to 1000 respondents as a result, this study encompassed 1000 respondents.“ – another strangely separated sentence; please capitalize the first letter of Required. In addition please clarify did you calculate the sample size for a proportion of 50% or any other or even a mean value? I am curious why the population of Riyadh is considered as unknown? It should be known.
„The validity and reliability were determined using a pilot study conducted among randomly selected HCUs (n=20)“ – the abbreviation HCU should be defined.
Please clarify how did you calculate the reliability, which measure did you use (there are plenty of them).
The sentence “The data was collected using a Convenience sampling procedure, with the help of a group leader from each college” is not clear. Please provide more detailed explanations about these leaders and the colleges.
The sentence “We used this sample strategy because it was easier to collect data and more convenient than other methods” is very strange. Yes, we all know that it is the easiest way to collect data. But let me highlight that we do not collect data just to know the opinion of the sample. In fact we want to know what the opinion of the population is. And the convenience sampling does not ensure the option for generalization of the results. They remain valid for the sample only. As researchers we should face the difficulties and if we cannot overcome them than we should describe them within the methodology and discussion sections.
The last sentence of the Methodology section should also start with a capital letter.
Data analysis
ANOVA is an abbreviation (no need to be defined) therefore all letters should be capitalized.
Nothing is mentioned about the descriptive statistics of the numerical variables. It is seen within the text that you presented them as mean and SD. But did you test them for normality? Please provide the test as well as its result. And if the variables are not normally distributed it is better to use median and interquartile range. In addition not-Gaussian variables could not be put in ANOVA or t-test comparisons. You should use non-parametric tests instead.
The age is another numerical variable among your dataset. So it is better to report it as average (mean or median) and dispersion measure (SD or IQR) in addition to the age intervals.
Results
I already mentioned for the age, so please add the average age to Table 1.
Table 2. Some issues are detected, instead of parenthesis there are numbers (see the attached file). The last sentence should be „AI allows the patients to increase their control over their health“
Page 7, the first paragraph: again if the score is not normally distributed you should use the median.
Table 4. I would suggest to add an additional analysis combining high school and university into one group and then compare them to lower educated respondents. In addition all p<0.1 but still >0.05 could be described as borderline significance (for the comparison between males and females) and also the text should be corrected. Your sentence influences that there is a significant difference but in fact its significance is borderline. Please add all the p-values in the text.
Discussion
The limitations of the study should start at a new line. Add some more limitations. For example the sample is convenient. You could compare the age and sex distribution of your respondents to the actual distribution among the study population – those 20,000 that you refer to calculate the sample size.

this study aims to Assessment
A total of 830 individuals completed and responded to the study questionnaires by giving a response rate of 83% (n=830). In this study majority of the respondents were male (69.4%), while 30.6% of them were female, 46% of them were above 41 years old, fallowed 20.8% of them were between the ages of 19 and 25 years old as shown in Table 1.
Author Response
Dear Authors,
Your article is very interesting! However, I have several comments:
Response: Thank you for the comment we appreciate it
The Abstract needs some improvement. The aim of the study is to assess (not to assessment). The statistical software and methods should not be mentioned there. The number of the respondents should be moved to methods. „This is a cross-sectional web-based questionnaire study that was conducted between January to April 2024 among 830 participants of which 69.4% were males, 46% of them were aged above 41 years old “. I would suggest to avoid +- sing when reporting SD. You could use parenthesis of comma instead. The sentence „Similarly, the mean perception score was higher among young adults (38.9±6.1) compared to another age group “is not clear. Did you mean the other age groups? And please clarify what do you mean by young (name the age group instead of describing it). Is this difference significant? I would suggest to add the p-values to all comparisons.
Response: Thank you so much for the comment, we have modified the abstract as suggested
The conclusion seems to consist of one sentence but strangely separated. Please rewrite it.
Response: We apologies for this, I have corrected the conclusion, as … The results showed that the Saudi public had a favorable opinion and perceptions of AI in healthcare. Suggesting to offer health management recommendations on how to successfully integrate and use medical AI while maintaining patient safety.
Methodology
„Eligible individuals were over the age of >18 years “– did you mean 18 and above? Please provide more details how did you ensure the participation of different types of individuals.
Response: Thank you for the comment, we have approached all the individuals, as much as we can, there is no specific criteria to approach adults with specific age, out inclusion criteria is that individuals must be aged above18 only eligible to fill the survey
I would also suggest to restructure the section: describe the study population; the sample size calculation; the respondents ‘selection; the questionnaire.
Response: Thank you for the comment, we have added each in separate paragraph as suggested
„The sample size for this study was calculated using the online calculator by considering an unknown (n=20,000) population of Riyadh in Saudi Arabia, at a 5% margin of error (ME) and a 95% confidence interval (CI). required sample was estimated to be 388, however to account for nonresponse and reduce data collecting errors we approached to 1000 respondents as a result, this study encompassed 1000 respondents. “ – another strangely separated sentence; please capitalize the first letter of Required. In addition please clarify did you calculate the sample size for a proportion of 50% or any other or even a mean value? I am curious why the population of Riyadh is considered as unknown? It should be known.
Response: Thank you for the comment, we apologies for the errors, we have corrected the sentence,
further we incorporated the total sample of Riyadh (n= 7821000) instead of unknown, and the required sample was 385 according to Raosoft, the detailed calculations are attached in the image
„The validity and reliability were determined using a pilot study conducted among randomly selected HCUs (n=20) “ – the abbreviation HCU should be defined.
Response, it was corrected as… The validity and reliability were determined using a pilot study conducted among randomly selected individuals (n=20) from Riyadh. The findings of the pilot study were not included in the main results and there were no amendments were made in the questionnaires. The reliability of the questionnaires was found to be 0.80, indicating the questionnaires are valid to use in the study.
Please clarify how did you calculate the reliability, which measure did you use (there are plenty of them).
Response, it was calculated using Cronbach alpha value
The sentence “The data was collected using a Convenience sampling procedure, with the help of a group leader from each college” is not clear. Please provide more detailed explanations about these leaders and the colleges.
Response: we apologise , we have corrected …The data was collected from individuals living in Riyadh region Saudi Arabia. For data collection, electronic questionnaires were prepared using Google form, and sent to the targeted population, using social media. In addition, before beginning the study questionnaires in the Google form, there was a statement about the significance and confidentiality of the data, who agreed to proceed would be redirected to the main questionnaires, and it was considered informed consent. Furthermore, the follow-up with the researcher, was done to achieve the maximum responses, who sent reminders to fill out the survey and send it back. The data was collected using online questionnaires which were handed to researcher from the college of pharmacy, clinical pharmacy department
The sentence “We used this sample strategy because it was easier to collect data and more convenient than other methods” is very strange. Yes, we all know that it is the easiest way to collect data. But let me highlight that we do not collect data just to know the opinion of the sample. In fact, we want to know what the opinion of the population is. And the convenience sampling does not ensure the option for generalization of the results. They remain valid for the sample only. As researchers we should face the difficulties and if we cannot overcome them than we should describe them within the methodology and discussion sections.
Response: thank you for the comment, we agreed that and therefore we removed that’s sentence
The last sentence of the Methodology section should also start with a capital letter.
Response: thank you for the comment, I have corrected to small letter we apologize for this error
Data analysis
ANOVA is an abbreviation (no need to be defined) therefore all letters should be capitalized.
Response: thank you for the comment, we have corrected
Nothing is mentioned about the descriptive statistics of the numerical variables. It is seen within the text that you presented them as mean and SD. But did you test them for normality? Please provide the test as well as its result. And if the variables are not normally distributed it is better to use median and interquartile range. In addition, not-Gaussian variables could not be put in ANOVA or t-test comparisons. You should use non-parametric tests instead.
Response: yes we added mean, std in analysis part , the data was normally distributed and we performed Shapiro–Wilk test, which was significant p=0.678
The age is another numerical variable among your dataset. So it is better to report it as average (mean or median) and dispersion measure (SD or IQR) in addition to the age intervals.
Response: thank you it is in categorical variable,
Results
I already mentioned for the age, so please add the average age to Table 1.
Response: thank you it is in categorical variable,
Table 2. Some issues are detected, instead of parenthesis there are numbers (see the attached file). The last sentence should be „AI allows the patients to increase their control over their health “
Response: it was by mistake and corrected
Page 7, the first paragraph: again if the score is not normally distributed you should use the median.
Response: thank you for the comment, however the data was normally distributed therefore we used parametric test
Table 4. I would suggest to add an additional analysis combining high school and university into one group and then compare them to lower educated respondents. In addition, all p<0.1 but still >0.05 could be described as borderline significance (for the comparison between males and females) and also the text should be corrected. Your sentence influences that there is a significant difference but in fact its significance is borderline. Please add all the p-values in the text.
Response: thank you for the comment, I have added the sentence of boarder line difference, however educational level, where only25 respondents had less than high school, comparing to highs cool and university, which significantly associated with means core
Discussion
The limitations of the study should start at a new line. Add some more limitations. For example, the sample is convenient. You could compare the age and sex distribution of your respondents to the actual distribution among the study population – those 20,000 that you refer to calculate the sample size.
Response: thank you for the comment, we have added the suggested limitations. ass The current study has several limitations. First, the data presented here were limited to one region in Saudi Arabia they may not fully reflect the knowledge of all Saudi adults in Saudi Arabia. Due to the self-administered nature of the questionnaire used to gather the data, recall or social desirability bias may have been present. Last the convenience sampling procedure, which might be one of the limitation to study. In addition, the age, and gender in the current study not exactly the number which used as references for sample size calculation, which suggest that it is recommended future research with a larger sample size, including data from other regions of Saudi Arabia is needed to better corroborate the findings
Comments on the Quality of English Language
this study aims to Assessment
A total of 830 individuals completed and responded to the study questionnaires by giving a response rate of 83% (n=830). In this study majority of the respondents were male (69.4%), while 30.6% of them were female, 46% of them were above 41 years old, fallowed 20.8% of them were between the ages of 19 and 25 years old as shown in Table 1.

Round 2
Reviewer 2 Report
Comments and Suggestions for Authors
Dear Authors,
Thank you for the efforts to improve the article! It is much better now. Some proofreading is needed (spaces missing before parentheses etc.)
Author Response
Responce: thank you for the comment, the entire manuscript was proofreaded